# Enhancing Leukemia Treatment: The Role of Combined Therapies Based on Amino Acid Starvation

**DOI:** 10.3390/cancers16061171

**Published:** 2024-03-16

**Authors:** Can Chen, Ji Zhang

**Affiliations:** 1Hermans B Wells Center for Pediatric Research, Indiana University School of Medicine, Indianapolis, IN 46032, USA; 2Department of Hematology, Affiliated Hangzhou First People’s Hospital, Westlake University School of Medicine, Hangzhou 310006, China

**Keywords:** leukemia, amino acid starvation, combination therapy, immunosuppression, GCN2

## Abstract

**Simple Summary:**

Targeting amino acid metabolism in leukemia therapy presents both opportunities and challenges. While disrupting amino acid utilization can hinder cancer cell growth and enhance treatment efficacy, achieving selective targeting to minimize damage to healthy cells is crucial. This review explores novel strategies for amino acid depletion-based treatments in leukemia, highlighting the potential of combining these approaches with traditional chemotherapeutics and immunotherapies to overcome resistance and improve patient outcomes.

**Abstract:**

Cancer cells demand amino acids beyond their usage as “building blocks” for protein synthesis. As a result, targeting amino acid acquisition and utilization has emerged as a pivotal strategy in cancer treatment. In the setting of leukemia therapy, compelling examples of targeting amino acid metabolism exist at both pre-clinical and clinical stages. This review focuses on summarizing novel insights into the metabolism of glutamine, asparagine, arginine, and tryptophan in leukemias, and providing a comprehensive discussion of perturbing their metabolism to improve the therapeutic outcomes. Certain amino acids, such as glutamine, play a vital role in the energy metabolism of cancer cells and the maintenance of redox balance, while others, such as arginine and tryptophan, contribute significantly to the immune microenvironment. Therefore, assessing the efficacy of targeting amino acid metabolism requires comprehensive strategies. Combining traditional chemotherapeutics with novel strategies to perturb amino acid metabolism is another way to improve the outcome in leukemia patients via overcoming chemo-resistance or promoting immunotherapy. In this review, we also discuss several ongoing or complete clinical trials, in which targeting amino acid metabolism is combined with other chemotherapeutics in treating leukemia.

## 1. Introduction

Amino acid depletion-based therapy has emerged as a new strategy in cancer treatment [1], especially in disrupting critical biosynthetic pathways in leukemic cells by limiting their access to amino acids or their downstream metabolites necessary for cellular function. Leukemic cells frequently rely on specific amino acids for growth, division, and survival. Targeting the metabolism of amino acids in tumors not only significantly inhibits leukemia progression but also plays a crucial role in overcoming drug resistance and bolstering the effectiveness of immunotherapy [2]. Pre-clinical studies have explored a variety of methods to induce amino acid deprivation, such as manipulating the concentrations of amino acids in cell culture media or utilizing drugs to block their cellular uptake and utilization [1,3,4,5]. This strategy aims to exploit the metabolic vulnerabilities of leukemic cells, which often display enhanced dependency on specific metabolic pathways when compared to normal cells.

Translating the concept of amino acid deprivation into a reliable clinical therapy for leukemia encounters several challenges. One challenge is how to kill leukemic cells without perturbing the function of normal cells to ensure safety and specificity in patients [6]. In addition, concerns of therapeutic resistance arise as leukemic cells may adapt to amino acid deprivation through multiple mechanisms, compromising the efficacy of the treatment [7]. Assessing the efficacy of amino acid deprivation requires comprehensive strategies. Combining treatments, such as pairing with other chemo-agents to enhance the potency of amino acid deprivation or combining with immunotherapy to overcome resistance, could significantly improve the outcomes in patients. This review focuses on novel insights into amino acid starvation-based therapies in leukemia and provides a comprehensive overview of combination strategies associated with the perturbation of amino acid acquisition and utilization in both pre-clinical and clinical settings.

## 2. Glutamine, a Versatile Precursor for Biosynthesis and Bioenergetics

### 2.1. Glutamine Catabolism and Its Regulation

Glutamine, one of the most abundant amino acids in the body [8], plays diverse roles in cellular metabolism, ranging from bioenergetics to the synthesis of nucleotides, glutathione (GSH), and various other non-essential amino acids [9]. Classified as a non-essential amino acid by itself, it can be synthesized de novo by using other carbon- and nitrogen-containing nutrients as precursors in mammalian cells [9]. It is also transported across cell and mitochondrial membranes by specific solute carrier family members, including SLC1A5, SLC38A2, SLC38A1, and SLC1A1 [10]. Its export occurs through the SLC7A5 antiporter in exchange for extracellular leucine [11].

The catabolism of glutamine initiates with its conversion to glutamate, which can be catalyzed by glutaminase (GLS) in many cancer cells [12,13], a process that also releases a free amide nitrogen molecule to biosynthetic pathways. Humans possess two types of glutaminase: kidney-type GLS1, distributed ubiquitously, and liver-type GLS2, primarily expressed in the liver [14]. In mammalian cells, the GLS1 gene encodes two isoforms of glutaminase: kidney-type glutaminase (KGA) and glutaminase C (GAC). Notably, the GAC isoform is predominantly found in acute myeloid leukemia (AML) [15]. Glutamate is subsequently converted by glutamate dehydrogenase (GDH) or by transaminases into α-ketoglutarate (α-KG). α-KG then can enter the tricarboxylic acid cycle (TCA), providing both energy and intermediates for macromolecule biosynthesis [16]. Remarkably, in certain cell types, glutamine serves as an equivalent energy source for ATP production as glucose [17]. Furthermore, glutamine is a precursor for nucleotides, is involved in the synthesis of other non-essential amino acids, and contributes to the formation of glutathione, a key cellular reducing equivalent to mitigate oxidative stress [18]. Indeed, many AML subtypes rely on producing glutathione to maintain intracellular redox balance [19], particularly in leukemia stem cells and residual leukemia cells after chemotherapy, suggesting vulnerability in redox homeostasis in these cells, which should be targeted to prevent leukemia recurrence or refractory disease [20].

Hematological malignancies heavily rely on glutamine for mitochondrial oxidative phosphorylation (OXPHOS), a vital energetic process to support cell survival and proliferation [15]. In leukemia, elevated expression of SLC1A5 facilitates glutamine uptake, a necessary process for the synthesis of glutathione and pyrimidine to drive chemo-resistance in the leukemic blast cells [21]. In an independent study, it was shown that the elevated expression of SLC1A5 is driven by the signal transducer and activator of transcription 3 (STAT3)–MYC protooncogene axis in AML stem and progenitor cells (LSCs), enhancing glutamine influx to support the TCA cycle and glutathione production [22].

### 2.2. Targeting Glutamine Metabolism in AML

Current strategies targeting glutamine metabolism in cancer cells focus on its depletion, glutaminase inhibition, and inhibition of glutamine uptake by blocking its membrane transporters. As a result, combined treatment strategies have emerged to combat drug resistance and enhance anti-tumor efficacy. AML cells are generally reliant on glutamine and mitochondrial OXPHOS for survival [15]. Notably, inhibitors of GLS1, such as BPTES, compound 938, and compound CB-839, are available, with CB-839 undergoing human clinical trials for hematological malignancies. Inhibiting glutamine metabolism via CB-839 significantly disrupts the production of glutathione in various AML types, leading to an accumulation of mitochondrial reactive oxygen species (mitoROS) and apoptotic cell death [15]. The inhibition of GLS1 induces apoptosis through caspase activation, predominantly involving the intrinsic mitochondrial apoptotic pathway [15]. In primary AML leukemic stem cells, high-level expression of BCL-2 supports OXPHOS, rendering the cells to be highly sensitive to BCL-2-specific inhibitors [23]. Combining CB-839 with one of these BCL-2 inhibitors, Venetoclax, demonstrates promising clinical potential for treating AML by inhibiting glutaminolysis, triggering mitochondrial depolarization, and inducing intrinsic caspase-dependent apoptosis, thus exhibiting potent anti-leukemic activity [15].

Combining glutaminase inhibition with agents that disrupt mitochondrial redox balance intensifies leukemic cells’ vulnerability to cell death. In a pre-clinical study, combining CB-839 with arsenic trioxide (ATO) or homoharringtonine (HHT), two known ROS inducers, significantly exacerbated cellular ROS production and cell death in AML cell lines in vitro and a secondary mouse AML model in vivo [24]. Furthermore, this combination therapy effectively eliminated primary AML colony-forming cells, indicating potential efficacy against leukemia stem cells. Notably, the CB-839 and HHT combination also demonstrates efficacy in treating acute lymphoblastic leukemia (ALL), another type of leukemia originating from lymphoid progenitor cells.

Mutations in the FLT3 receptor type-III kinase are prevalent in about 30% of AML patients, primarily characterized by FLT3 internal tandem duplication (ITD) mutations. These mutations represent an increased rate of relapse post-standard therapies and a poor prognosis [25]. Investigation by metabolomics and gene-expression profiling analyses revealed that in FLT3-ITD, AMLs specifically depend on glutamine metabolism during treatment targeting the tyrosine kinase activities of the FLT3 receptors. Inhibition of FLT3-ITD in these AMLs disrupts glutamine metabolism by blocking its uptake via an undefined mechanism. As a result, inhibition of glutamine uptake depletes glutathione and induces apoptosis due to increased mitochondrial oxidative stress [26,27]. Combining AC220, a FLT3 inhibitor, with CB-839 further depletes glutathione, induces mitochondrial ROS, and triggers apoptotic cell death synergistically [27,28]. In vivo experiments using CB-839 demonstrated enhanced elimination of leukemic cells when AC220 was given simultaneously, significantly improving survival in a patient-derived xenograft AML model [27]. Similarly, the combination of imatinib, a BCR/ABL inhibitor, with CB-839 amplifies apoptosis in BCR-ABL-positive cells, characterized by increased intracellular ROS production and reduced oxygen consumption [28]. The completed clinical trial (Table 1), registered as NCT02071927, utilized a combination of CB-839 and azacytidine, a DNA methyltransferase (DNMT) inhibitor known for its demethylating effects in vitro and in vivo. Until now, no results from this trial have been made public. However, a similar combination has demonstrated efficacy in patients with myelodysplastic syndromes, as evidenced by the study registered as NCT03047993 (Table 1). In this trial, an objective response, defined as complete remission (CR), morphologic CR, or hematologic improvement, was achieved in 15 out of 23 patients, accounting for 65.2% of the participants [29].

### 2.3. Combination Therapeutics Involving Glutamine Perturbation and Traditional Chemo-Agents 

There is no drug that can deplete glutamine in circulation specifically till today. However, L-asparaginase, a chemo-agent that is used in ALL patients via degrading asparagine to aspartate and ammonia has caught our attention because L-asparaginase can also degrade glutamine but as a less effective substrate [33]. Indeed, a clinical study using crisantaspase, an asparaginase purified from Erwinia chrysanthemi, showed responses in relapsed AML patients, which correlated with its capacity to deplete glutamine in the plasma of these patients [34]. Pegcrisantaspase (PegC), a long-acting form of crisantaspase, effectively depletes glutamine and asparagine, and inhibits mRNA translation and cellular protein synthesis, leading to cell death in leukemic cells in vitro [35]. The combination of Venetoclax, a BCL2 inhibitor, with PegC enhanced the interaction between eIF4E and 4EBP1 within the cap-binding complex to inhibit global mRNA translation, which reduced the expression of MCL-1 protein and caused cell death in vitro and synergistically inhibited leukemia progression in a PDX AML model in vivo. In this study, it was shown that glutamine depletion by PegC played a major role in this process [35]. A clinical trial focusing on relapsed/refractory acute myeloid leukemia (R/R AML) is currently ongoing, investigating the combination of PegC and Venetoclax (NCT04666649, Table 1).

Residual AML cells exhibit transient metabolic adaptation that contributes to their chemo-resistance. Studies in mouse AML models suggest that strategically manipulating specific metabolic pathways, such as glutamine catabolism and pyrimidine synthesis, holds therapeutic promise [21]. Glutamine analogs and antagonists, such as 6-diazo-5-oxo-L-norleucine (DON), inhibit a spectrum of enzymes that utilize glutamine as substrates [36]. Administering DON concurrently with chemotherapy (CT) showed no difference in survival compared to CT alone. However, when DON treatment followed CT and covered the phase of maximal response, a distinct survival benefit was observed. Short-term DON treatment post-CT notably enhanced AML cell elimination, evident by increased apoptotic cells and higher instances of double-stranded DNA breaks in residual cells [21]. While DON was tested in clinical trials for both solid tumors and leukemia since the 1980s, it was however abandoned due to its limited efficacy as a single agent and the occurrence of dose-dependent side effects [36].

Targeting glutamine metabolism in leukemia aims to reduce its uptake, block its catabolism, and thereby prevent its further utilization (Figure 1). In recent work, a non-specific inhibitor of glutamine transporter SLC1A5, namely L-c-glutamyl-p-nitroanilide (GPNA), showed efficacy in inhibiting leukemia progression in a xenograft AML model in vivo [37]. It will be interesting in the future to test more specific inhibitors, such as V-9302, which has been found to be effective in solid tumor models [38]. Notably, inhibitors targeting FLT3-ITD show promise in inhibiting glutamine uptake and exhibiting synergistic effects with glutaminase inhibitors. Given the importance of GSH, a vital antioxidant produced via glutamine catabolism, in mitigating oxidative-stress-induced cell death, combining glutamine inhibition with other chemotherapy drugs that induce ROS seems to be a rational choice. Glutamine deprivation might overcome resistance to BCL-2 inhibitors by perturbing mitochondrial integrity and function. In the future, it warrants further investigation of other glutamine-dependent biosynthetic pathways, such as nucleotide biosynthesis, to determine its potential synergy with other chemotherapeutics in leukemia.

## 3. Asparagine and Its Depletion-Based Therapy in ALL Patients

### 3.1. Asparagine Metabolism and the Regulation of Its Biosynthesis

Asparagine is a non-essential amino acid in normal human cells due to their capacity to synthesize it from glutamine and aspartate via asparagine synthetase (ASNS) [39]. In contrast to the other 19 proteinogenic amino acids, asparagine catabolism in mammalian cells is not evident. Thus, it is believed that asparagine is only used as a precursor for protein synthesis in mammals [39]. Additionally, asparagine can be used as an amino acid exchange factor, regulating the uptake of arginine, histidine, and serine, which further coordinates protein synthesis and nucleotide synthesis by modulating mTORC1 activity [40]. Indeed, a recent work suggests that asparagine can directly regulate mTORC1 activity through a Rag GTPase-independent mechanism [41].

ALL cells are compromised in their ability to synthesize asparagine due to the low expression of ASNS [42]. Consequently, ALL cells rely heavily on circulating asparagine for growth and survival. AML cells display lower sensitivity to asparagine depletion when compared to ALL cells, likely due to their elevated expression of ASNS [43]. Currently, L-asparaginase stands as the sole clinically used agent targeting asparagine acquisition [44]. Its primary anti-leukemia effect involves hydrolyzing serum L-asparagine into aspartate and ammonia, thereby inhibiting protein synthesis [45]. Asparagine depletion by L-asparaginase directly impacts leukemic cells, affecting levels of ROS production, cell cycle progression, autophagy, and triggering apoptotic cell death [46]. In addition, in chronic myeloid leukemia (CML) cells, L-asparaginase treatment can inhibit the AKT/mTOR and ERK signaling pathways, contributing to its anti-tumor effect [47].

### 3.2. Application of L-Asparaginase and Its Potential Synergy with Other Therapeutics in Leukemias 

The efficacy of asparaginase-based regimens in pediatric ALL prompts trials of pediatric-inspired protocols with asparaginase for adolescent, young adult (AYA), and adult populations [48]. Therefore, in contrast to therapeutic approaches that deplete other amino acids, the regimen containing asparaginase and other chemo-agents has been widely investigated and deliberated in the context of ALL. In the setting of asparagine deprivation, the survival of leukemic cells relies on the expression of ASNS at least in cell culture, influencing their sensitivity to the treatment [33,39]. ALL cells adapt to asparagine depletion by activating the GCN2/eIF2α/ATF4 axis, leading to an increase in ATF4-dependent transcription of the ASNS gene [49,50]. Thus, inhibiting the GCN2 pathway may overcome resistance to L-asparaginase treatment. Indeed, GCN2 inhibitors render ALL cells to be more sensitive to L-asparaginase treatment by inhibiting ASNS induction, consequently reducing global protein synthesis and triggering apoptosis, likely through stress-activated MAP kinase pathways [50]. It is worthy of note that in contrast to cell culture studies, conflicting results exist regarding whether the expression of ASNS predicts clinical responses to L-asparaginase treatment. In a small cohort of B-ALL patients with TEL/AML1 fusion, no correlation was found in overall therapeutic responses and their capacity to induce the mRNA expression of ASNS following L-asparaginase treatment [51]. In contrast, in a larger cohort of T-ALL patients, low expression of ASNS mRNA in TLX1-positive T-ALL patients showed better responses to L-asparaginase treatment [52]. Therefore, whether combining GCN2 inhibitors and L-asparaginase has better therapeutic efficacy warrants further investigation in a clinical setting. 

In addition to the GCN2/ATF4 axis, the expression of ASNS gene also depends on the methylation status of the gene promoter [53,54]. The DNA demethylating agent decitabine was shown to induce the expression of ASNS in cell lines that express low levels of ASNS [54]. Despite the fact that decitabine treatment increases the expression of ASNS gene in these asparagine auxotrophic cell lines, a pronounced synergistic effect was observed when decitabine and L-asparaginase were used together in T-ALL cell lines in vitro [55]. This result indicates that the expression of ASNS is unlikely the explanation of the synergy observed in the study, which warrants further investigation to fully understand the underlying mechanisms.

Enhancing sensitivity to L-asparaginase is a key strategy in ALL treatment. One approach involves a β-catenin-independent branch of the WNT signaling, known as WNT-dependent stabilization of proteins, which inhibits glycogen synthase kinase 3 (GSK3)-dependent protein degradation, a process recycling amino acids for cellular usage [56]. It has been shown that pharmacological inhibition of GSK3 profoundly sensitizes drug-resistant leukemias to asparaginase [57,58]. Additionally, inhibition of the Bruton’s tyrosine kinase (BTK) signaling pathway interferes with c-MYC-induced proteotoxic stress and thereby suppresses GCN2-dependent AAR pathways [59,60]. As a result, ibrutinib, a BTK inhibitor, suppresses GCN2 activities and synergizes strongly with L-asparaginase in B-ALL, which is currently undergoing clinical trials [60]. Of interest, a recent work suggests that asparagine starvation can directly inhibit the translation of c-MYC mRNA in leukemia/lymphoma cells [61]. In a mouse B-cell lymphoma model, they showed that combining L-asparaginase with phenformin, an electron transferring chain (ETC) inhibitor that depletes intracellular aspartate, an indispensable precursor for asparagine, significantly reduced c-MYC expression and lymphoma burden [61]. Moreover, the gene encoding the system L amino-acid transporter 1 (LAT1), termed SLC7A5, is overexpressed in ALL and many other proliferating cells [62,63]. Inhibiting LAT1 with a potent and selective inhibitor (JPH203) decreases leukemic cell viability and proliferation, which can be further exacerbated when L-asparaginase was given simultaneously to engage an unfolded protein response (UPR) and cell death [62]. 

In clinical settings, apart from its combination with traditional chemotherapy, L-asparaginase has been used with other targeted drugs too. For instance, blinatumomab, a bi-specific T cell engager to recognize the CD19 antigen on the B cell surface, has shown activity in refractory/relapsed (R/R) BCP-ALL. When used in combination with VANDA (etoposide, cytarabine, mitoxantrone, dexamethasone, and asparaginase), it achieved complete remission in a significant proportion of patients [64]. While in a case report, combining inotuzumab ozogamicin (anti-CD22) with high-dose methotrexate and pegaspargase in a patient with extramedullary B-ALL resulted in toxic liver damage [65]. In the EWALL-PH-01 study (Table 1) about tyrosine kinase inhibitors (TKIs) in managing Ph-positive ALL, administration of dasatinib with a low-intensity asparaginase-containing regimen in older patients resulted in favorable outcomes [31]. In contrast, significant toxicity was noted in the UKALL14 study (Table 1) using imatinib instead [32].

Despite its effectiveness, asparaginase has notable toxicities, including hepatotoxicity, thrombosis, immunosuppression, and pancreatitis [66]. It was shown that the hepatotoxicity during asparaginase treatment is associated with amplified amino acid response (AAR), which is exacerbated in GCN2-deficient mice [67]. Indeed, GCN2-dependent regulation of fibroblast growth factor 21 (FGF21) and antioxidative defenses influences the liver’s adaptability during prolonged treatment. Loss of this regulation affects lipid metabolism and liver homeostasis detrimentally [68]. In addition, GCN2 loss predisposes the pancreas to maladaptive endoplasmic reticulum (ER) stress responses and autophagy during asparaginase treatment, potentially contributing to asparaginase-associated pancreatitis [69].

ASNS plays a critical role in driving ALL cells’ resistance to L-asparaginase treatment. Activation of the GCN2-ATF4 pathway is a pivotal step to enhance the expression of ASNS. Therefore, targeting GCN2 or asparagine biosynthesis itself alongside L-asparaginase holds therapeutic promise. Combined targeted therapeutic approaches based on crosstalk between signaling pathways have shown significant effects in pre-clinical studies (Figure 2), such as combining GSK or BTK signaling inhibition to overcome L-asparaginase resistance. Bi-specific antibodies are extensively used in leukemia treatment and have shown clinical efficacy when combined with L-asparaginase too, broadening the applications of asparagine depletion in treating leukemia. In the future, further studies are needed to determine the mechanistic insights of new combination therapeutics involving L-asparaginase, as well as potential toxicity and side effect, for better management of patients in clinical settings.

## 4. Arginine Deprivation: Anti-Leukemic or Immuno-Suppressive?

### 4.1. Arginine Metabolism and Its Depletion-Based Therapy in AML 

Arginine is also a non-essential amino acid and plays a pivotal role in the human body. Arginine is synthesized both in the liver and in the kidneys through the urea cycle [70]. A critical precursor to drive the urea cycle is ornithine, which can be produced from the catabolism of glutamate or proline. The entire urea cycle requires metabolic intermediates to cross the mitochondrial membrane twice and involves three important reactions catalyzed by ornithine transcarbamylase (OTC), argininosuccinate synthase (ASS1), and argininosuccinatelyase (ASL) [70]. Arginine is used to synthesize protein, and to generate urea and agmatine, in normal cellular processes. Cells acquire arginine externally through various cationic amino acid transporters (CATs), notably CAT-1 and CAT-2B, due to their high affinity for this amino acid [71]. While the human body can produce arginine de novo, dietary intake remains the primary source for its acquisition, indicating a potential limitation in its de novo biosynthesis [72]. 

The catabolism of arginine involves four enzymes: nitric oxide synthase (NOS), arginase (ARG), arginine decarboxylase (ADC), and arginine:glycine amidinotransferase (AGAT) [70]. NOS and ARG compete for arginine utilization: NOS breaks it down into citrulline and nitric oxide (NO), while ARG converts it into ornithine and urea. ADC converts arginine into agmatine, although its biochemical activities were studied mainly in plants, bacteria, and invertebrates. In contrast, AGAT converts arginine into ornithine and creatine within the mitochondria [73]. Creatine is phosphorylated by creatine kinase to form creatine phosphate, which can be used to generate ATP during short bursts of intensive exercise. Creatine phosphate can spontaneously cyclize to form creatinine, which is the final product of the arginine–creatine pathway [74]. In AML blasts, both CAT-1 and CAT-2B expression is consistent across blast subtypes, while in ALL blasts, CAT-1 is predominantly expressed [75,76]. Critical enzymes in the biosynthesis of arginine, such as ASS1 and ASL, are notably missing in AML and ALL, rendering these cells to heavily rely on extracellular arginine availability [76,77].

Strategies to reduce arginine concentrations in the microenvironment of arginine auxotrophic tumors include dietary arginine restriction, inhibiting cell surface transporters, and using arginine-depleting enzymes, such as arginine decarboxylase (ADC), arginine deiminase (ADI), and arginase, to deplete extracellular arginine [78]. Among these approaches, arginase is the most used. BCT-100, a clinical-grade pegylated recombinant human arginase, catalyzes arginine conversion to ornithine and urea, effectively depleting arginine from circulation [79]. In a murine xenograft model of ALL, BCT-100 lowers plasma arginine levels and reduces ALL blast counts [75]. Cytarabine, a key agent in AML chemotherapy, inhibits DNA polymerases by incorporating into DNA strands, leading to S-phase arrest and cell death [80]. When combined with BCT-100, cytarabine showed greater cytotoxicity in AML samples than either compound alone, probably via targeting different phases of the cell cycle [76]. Similarly, in a pre-clinical study of ALL, the combination of pegylated arginase I (peg-Arg1) with cytarabine significantly inhibited T-ALL progression and extended overall survival in mice when compared to peg-Arg1 alone [77]. Another study using primary ALL blast cells showed that a combination of BCT-100 with dexamethasone exhibited synergistic cytotoxicity than BCT-100 alone [75]. A clinical trial was conducted to evaluate the efficacy of PEG-BCT-100, a pegylated arginine deiminase, in patients with relapsed or refractory acute myeloid leukemia (AML) in terms of remission rate (NCT02899286, Table 1). As of now, the results of the trial have not been published or made available.

The pegylated human recombinant arginase I cobalt (HuArg1(Co)) exhibits significantly enhanced serum stability compared to native human L-arginase I [81]. In AML, combining HuArgI(Co)-PEG5000 with cytarabine results in lower IC50 values, ranging from 2- to 18-fold, than when HuArgI(Co)-PEG5000 is used alone. Similarly, the combination of HuArgI(Co)-PEG5000 with doxorubicin produces IC50 values that are 2.5- to 5-fold lower than the values observed with each agent individually. AML cells exhibit arginine auxotrophy and are sensitive to arginine depletion, with all tested AML cell lines showing sensitivity to HuArgI(Co)-PEG5000. Conversely, normal hematological cells, such as CD34+ bone marrow progenitor blasts, display resistance to HuArgI(Co)-PEG5000, highlighting the selectivity of arginine deprivation-induced cytotoxicity towards AML cells. Furthermore, the combination of HuArgI(Co)-PEG5000 with standard AML chemotherapeutic agents, such as cytarabine and doxorubicin, demonstrates potential additive or synergistic effects in a subset of leukemia cell lines [82]. Two clinical studies were conducted on PEG5000 in treating AML (Table 1). However, one study was terminated (NCT01551628) due to slow patient enrollment, while the other has not yet released any results (NCT02732184).

Arginine deiminase (ADI), initially isolated from Pseudomonas putida, was first used to treat murine leukemia lymphoblasts as an anti-tumor agent in vitro; however, its effectiveness was not observed in vivo [83]. Later development of ADI, purified from Mycoplasma, successfully inhibited the growth of several murine and human tumor cell lines both in vitro and in vivo [84]. Further studies demonstrated the efficacy of ADI in treating melanomas and hepatocellular carcinomas (HCCs) in mice [85]. The broader application of ADI as a standard anti-cancer treatment remains limited due to its low efficiency, which is primarily attributed to its antigenicity in mammals—as it is a microbial enzyme—and its short serum half-life, necessitating daily injections in mice at least [85]. To overcome these limitations, a polyethylene glycol (PEG)-formulated ADI, was developed, namely ADI-PEG-20, which increases its stability and reduces the required dosages during treatment. This formulation has been shown to retain the in vitro effectiveness of native ADI against melanomas and HCCs, while significantly improving its in vivo efficacy [86]. In addition, ADI has been found to induce cell cycle arrest and apoptosis in human leukemia cells [87], highlighting its potential as a therapeutic agent against a range of arginine-dependent cancers. ADI-PEG-20 is undergoing clinical trials in ASS1-negative solid tumors [88]. While the use of ADI-PEG 20 as a single agent in treating relapsed/refractory acute myeloid leukemia (R/R AML) showed unfavorable results (NCT01910012), even in patients with ASS1 deficiency [89], a combination therapy for high-risk AML with Venetoclax and Azacitidine is currently ongoing (NCT05001828, Table 1). In AML mouse models, a synergistic effect between cytarabine and ADI-PEG-20 was demonstrated in vivo. The combination of ADI-PEG 20 and cytarabine demonstrated superior efficacy compared to either treatment alone, achieving positive responses in all six acute myeloid leukemia (AML) cases tested in xenograft models [90].

### 4.2. Arginine Supports the Anti-Tumor Effect of T Cells 

Although arginine depletion exhibits anti-leukemia activity, it also leads to an exhaustion of arginine in the microenvironment. In AML blasts, arginine catabolism is primarily driven by arginase (ARG). Elevated expression of either ARG1 or ARG2 can contribute to arginine depletion in the circulation and T cell suppression, although the mechanisms of their upregulation might be different [91,92]. These blasts lack a complete set of arginine-recycling enzymes, underscoring their dependence on extracellular sources of arginine [76]. The augmented metabolic demand for arginine in activated T cells is evident, with arginine further promoting T cell survival and activation via BAZ1B, PSIP1, and TSN signaling pathways [93]. In addition, both germline deletion of the Arg2 gene in mice and the adoptive transfer of Arg2-deficient CD8+ T cells significantly improve the anti-tumor effect in pre-clinical cancer models. This enhanced anti-tumor effect is linked to enhanced activation, effector functionality, and the sustained presence of CD8+ T cells where an arginine catabolic enzyme is missing [94].

Of interest, it was reported that patient-derived AML blasts are able to suppress T cell proliferation via the secretion of ARG2, which directly degrades the circulating arginine needed for T cell activation and functionality [91]. The immunosuppressive activity in the plasma derived from AML patients can be mitigated through the use of small-molecule inhibitors of ARG2 (NOHA), or by supplementing with high levels of arginine [91], direct evidence for the involvement of ARG activity in the circulation to drive immunosuppression. Indeed, significantly lower circulating arginine concentrations in AML patients were found when compared to healthy controls, which have notable impacts on T cell function and chimeric antigen receptor (CAR)-T cell cytotoxicity [91,95]. Thus, inhibiting arginine catabolism in AML patients may enhance CAR-T cell cytotoxicity against AML cells. In the future, it is critical to leverage the circulating arginine levels to ensure adequate suppression of AML blast proliferation while minimizing its effect on immunosuppression. 

Tumor cells often do not express all biosynthetic enzymes that are needed for de novo biosynthesis of arginine. As a result, they often excessively consume arginine from the tumor microenvironment, leading to arginine depletion and subsequent immunosuppression. Thus, targeting arginine biosynthesis in tumor cells may cause further consumption in the tumor microenvironment, and strategies like combined cell-based immunotherapy need to consider their impacts on different cell types (Figure 3). Combined therapies targeting tumor cells mainly focus on using recombinant ADI or ARG to deplete arginine in combination with other chemotherapies, validated for efficacy at least in cell culture experiments. In the setting of CAR-T cell therapy, strategies targeting arginine availability in the tumor cells should also consider replenishing arginine in the microenvironment to preserve CAR-T activity. Future research should also optimize the timing of arginine depletion and immunotherapy to achieve a persistent anti-leukemic effect.

## 5. Tryptophan, an Immune Modulator

### 5.1. Tryptophan Catabolic Enzymes and Their Roles in Immune Cells in the Tumor Microenvironment

Tryptophan (Trp) is an essential amino acid obtained from our diet. Cells predominantly acquire tryptophan through the widely expressed neutral amino acid transporter system L [96]. Although a small portion of free tryptophan contributes to protein synthesis and the production of neurotransmitters like serotonin and neuromodulators, such as tryptamine 2, more than 95% of tryptophan undergoes degradation via the kynurenine (Kyn) pathway [97]. This pathway generates diverse metabolites that significantly impact immune responses. The rate-limiting step within the Kyn pathway is the enzymatic conversion of tryptophan to N-formylkynurenine (NFK) mediated by indoleamine-2,3-dioxygenase 1/2 (IDO1/2) and tryptophan-2,3-dioxygenase (TDO). Depletion of tryptophan by these enzymes profoundly affects the survival and functions of immune cells [98]. In hematopoietic malignancies, the expression of IDO1 in the leukemic cells emerges as the primary enzyme involved in tryptophan catabolism [99].

Tryptophan depletion and the consequent activation of the aryl hydrocarbon receptor (AhR) by kynurenine within the tumor microenvironment (TME) play a critical role in reshaping the interaction between immune and leukemic cells, thereby promoting immune surveillance and leukemia progression. The Kyn pathway produces metabolites, like kynurenine, that act as endogenous ligands for AhR, leading to an activation of PD-1 expression on T cell surface to induce immune tolerance [100]. In addition, AhR activation can induce the differentiation of naïve T cells into regulatory T cells (Tregs) and suppress the development of cytotoxic T lymphocytes (CTLs), which helps leukemia cells evade immune detection [101]. Furthermore, AhR signaling induces a tolerogenic phenotype in dendritic cells (DCs), diminishing their antigen-presenting capabilities and production of pro-inflammatory cytokines, thereby impairing the initiation of effective anti-leukemia immune responses [102]. Finally, AhR activation strengthens the immunosuppressive functions of myeloid-derived suppressor cells (MDSCs), frequently observed in the microenvironment of leukemic cells, further suppressing T cell responses and facilitating leukemia progression [103]. Collectively, the activation of AhR by tryptophan catabolites plays a critical role in the TME to dampen immune responses, which contributes to immune evasion and leukemia progression.

Leukemic cells often exhibit an elevated expression of IDO1, accelerating tryptophan breakdown, which correlates with a poor prognosis in patients [99,101]. One possible mechanism of IDO-mediated immunosuppressive effect is likely through tryptophan depletion in the microenvironment of leukemic cells, similar to what has been described in solid tumors [104]. Tryptophan depletion triggers a cellular stress response, leading to G1 phase cell cycle arrest in T cells. This response relies on the GCN2 pathway, activated by an increase in uncharged tryptophan-tRNA in T cells [105]. Another mechanism that IDO can contribute to immune suppression is through the production of metabolites in the kynurenine pathway, such as 3-hydroxykynurenine and 3-hydroxyanthranilic acid, both of which have been shown to play a role in suppressing T cell activation and proliferation [106]. Indeed, many metabolites resulting from tryptophan degradation have been linked to a potential trigger of apoptosis in T cells [107]. Understanding the connections between IDO activity, its metabolites, and their effects on T cells could pave the way for combination therapeutic approaches in leukemia. Specifically, combining IDO-targeted strategies with immunotherapies like adoptive T cell transfer and CAR-T therapy holds promise as a potential treatment avenue.

### 5.2. Pre-Clinical and Clinical Applications of IDO Inhibitors in Leukemia

The IDO inhibitor 1-methyl tryptophan (1-MT) stands as one of the extensively studied drugs in this field. Other inhibitors like epacadostat and indoximod are currently undergoing clinical trials [104,108]. Donor leukocyte infusion (DLI), a cellular immunotherapy involving infusions of lymphocytes from donors, is utilized to induce stable remission in patients with post-transplant leukemia relapse [109]. However, a challenge with DLI is the risk of uncontrolled graft-versus-host disease (GVHD), which can lead to severe complications and, in some cases, mortality. Despite DLI procedures, relapses can still occur. Thus, combining an immunomodulatory agent with DLI after reduced-intensity allo-SCT (allogeneic stem cell transplantation) emerges as an appealing strategy from a clinical standpoint. Studies in a mice model of graft-versus-leukemia (GVL) have shown that both stereoisomers of 1-MT enhance DLI-induced GVL effects without exacerbating GVHD, potentially through the inhibition of IDO1 activity [110]. In the realm of B-cell malignancies, CD19-specific CAR-T cells offer promising therapeutic potential [111]. However, tumor IDO activity can impede CD19-CAR-T therapy by acting through tryptophan catabolism. The combination treatment of CAR-T and 1-MT has shown significantly improved tumor control compared to either therapy alone. The IDO inhibitor (1-MT) acts to shield CAR-T cells from the detrimental effects exerted by IDO-positive tumors [112]. 

The combination of 1-MT with chemotherapy also demonstrated noteworthy results. There was a significant suppression in the proliferation of AML blast cells when doxorubicin was used alone, and this suppression was even more pronounced when 1-MT was used together [113], suggesting a synergistic effect between these two drugs. Tumor cells release a burst of tumor-associated antigens that can enter the antigen-presentation pathway during chemotherapy-induced cell death. However, most chemotherapeutic agents do not seem to elicit a detectable protective immune response against established human tumors. This situation is partly because the tumor antigens released by chemotherapy are presented in tumor-draining lymph nodes, where studies in mice have shown that IDO expression can turn these nodes into an environment that promotes immunosuppression and tolerance induction [114,115]. Therefore, inhibiting IDO in the period following chemotherapy could potentially delay or interfere with the re-establishment of tolerance to tumor antigens [108]. This synergy between IDO inhibitors and chemotherapy might be explained by the ability of IDO inhibitors to help the immune system eradicate residual cancer cells after chemotherapy [116]. 

The L-isomer of 1-MT functions as a weak substrate for the enzyme IDO1 and accounts for the mild inhibitory effect observed with the racemic mixture on this enzyme. On the other hand, the D-isomer does not interact with or inhibit the IDO1 enzyme in its purified form [117]. D-1-MT inhibits the IDO pathway without possessing direct enzymatic inhibition against IDO1. Although it is less understood, the effect of D-1-MT may rely on its preferential inhibition of IDO2 [118], another enzyme related to the tryptophan catabolism in tumors and tumor-draining lymph nodes. This selective inhibition could explain the paradoxical observation that D-1-MT empowers chemotherapy and relieves T cell suppression despite not directly inhibiting IDO1 enzymatic activity [119]. Despite this, clinical development has concentrated on the D-isomer of 1-MT, now known as indoximod, driven by pre-clinical evidence suggesting its superior anti-tumor efficacy and unique mechanisms of action [117]. In an open-label, multicenter, phase 1 study (NCT02835729, Table 1), newly diagnosed AML patients eligible for treatment received indoximod in combination with induction chemotherapy (idarubicin with cytarabine). Among 25 intention-to-treat patients, 21 (84%) achieved remission (CR/CRh/Cri/CRp). Of the 12 patients with measurable residual disease (MRD) data available during remission, 10 (83%) reached MRD levels below 0.02% (MRD-neg). The combination of indoximod with standard AML induction therapy was well tolerated [30]. These results underscore promising outcomes, showing encouraging rates of both morphologic response and attainment of MRD-negative status within this patient cohort, although there is no evidence thus far whether the promising outcome is due to restoration of immune responses.

Cell immunotherapy is currently a hotspot in cancer treatment. Tryptophan depletion and/or its catabolic products play crucial roles in immune suppression, involving pathways like GCN2 and AhR. Therefore, strategies targeting tryptophan mainly focus on inhibiting the key enzyme IDO in this metabolic pathway to preserve immune response (Figure 4). Clinical data have confirmed the advantage of combining IDO inhibitors with chemotherapy in treating AML. In addition, targeting IDO can enhance the effectiveness of CAR-T or adoptive immunotherapy, suggesting potential benefits for patients facing relapse/refractory cases, or undergoing allogeneic hematopoietic stem cell transplantation, warranting further clinical exploration for safety and efficacy. Furthermore, the mechanism that regulates IDO1 expression in leukemic cells or how the kynurenine pathway suppresses T cell function is still not fully understood. Exploration of these new directions will facilitate the development of other targeted therapeutic strategies to treat leukemia by manipulating tryptophan metabolism in both leukemic cells and T cells.

## 6. Conclusions

Perturbing amino acid metabolism has proven effective in treating leukemia. Various strategies have been developed based on the essential role of amino acids in leukemic cells, as well in as immune cells. In summary, leukemic cells require substantial amino acid uptake to meet their high biosynthetic demand. Thus, blocking amino acid uptake and further utilization have shown efficacy in both pre-clinical and clinical research. On the other hand, depletion or catabolism of arginine or tryptophan may cause an immunosuppressive effect to compromise immunotherapy. Similar scenarios have been observed with other amino acids, including glutamine and asparagine [120,121,122], which can lead to immunosuppression or activation in certain circumstances. Future studies are needed to achieve a balance between adequate anti-leukemic effect and a persistent immune-responsive microenvironment. 

## Figures and Tables

**Figure 1 cancers-16-01171-f001:**
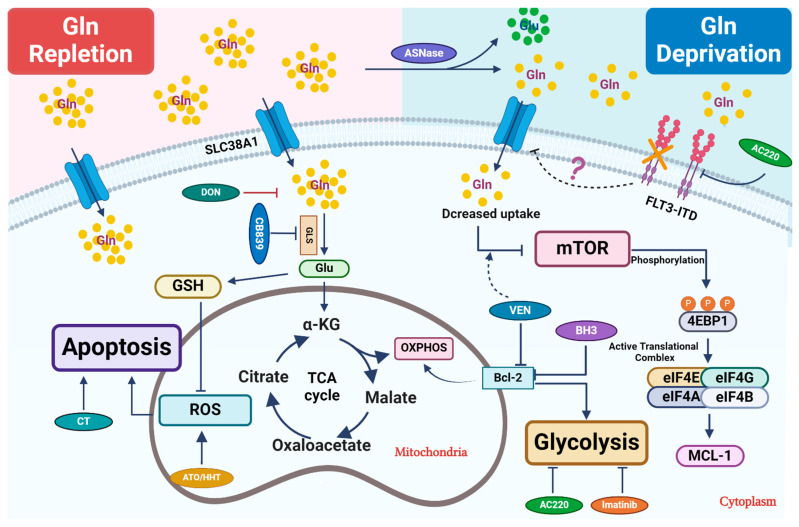
Therapeutic strategies that perturb glutamine metabolism. Glutamine (Gln) is transported into cells through cell surface transporters (SLC38A1). It is used as a precursor for protein synthesis. Intracellularly, glutamine is broken down into glutamate (Glu) by enzymes like GLS. Glutamate enters the TCA cycle via further catabolism into a-ketoglutarate (α-KG), which is critical to replenish the TCA cycle intermediates for other biosynthetic pathways or ATP production through oxidative phosphorylation (OXPHOS). Glu is also a critical precursor to synthesize glutathione (GSH), a key antioxidant to mitigate oxidative stress in leukemic cells. Combined treatment primarily focuses on blocking glutamine uptake or blocking its conversion into glutamate. In certain settings, L-asparaginase (ASNase) can deplete extracellular glutamine, which further reduces its intracellular utilization. Venetoclax (VEN) enhances the inhibitory effect of amino acid deprivation caused by ASNase treatment on mTOR, ultimately leading to the suppression of MCL-1 protein expression. 6-diazo-5-oxo-L-norleucine, also known as DON, is a glutamine analog. It disrupts multiple metabolic reactions that utilize glutamine as a substrate. CB-839, a GLS inhibitor, inhibits glutamate production and significantly increases ROS when combined with oxidizing agents, such as ATO/HHT. BCL-2 enhances mitochondrial OXPHOS for energy production, and thus its inhibition synergizes with CB-839 to disrupt energy production and mitochondrial function. Additionally, FLT3-TKI (AC220) reduces glutamine uptake and utilization in leukemic cells and thus shows synergy with CB-839, likely through perturbing bioenergetics.

**Figure 2 cancers-16-01171-f002:**
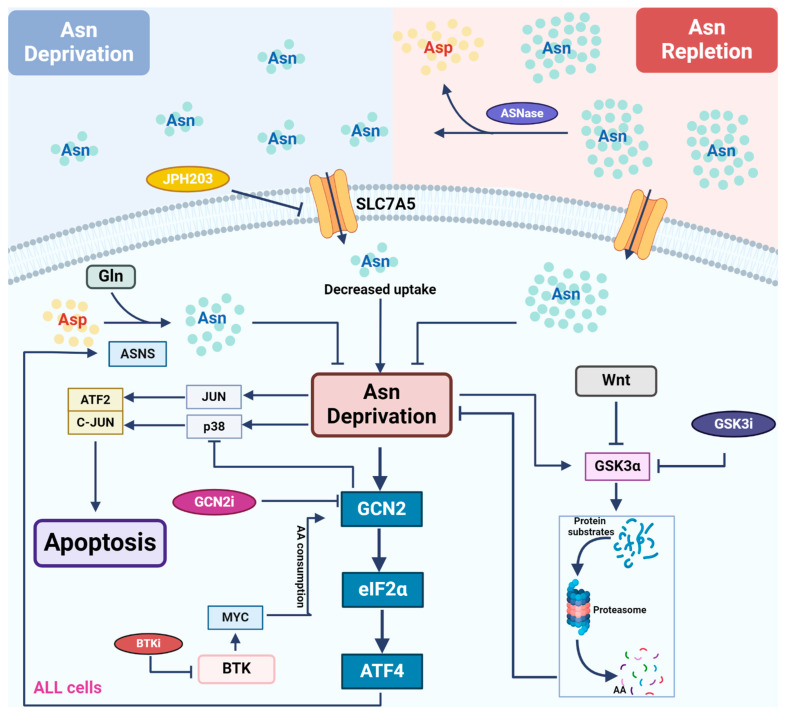
Asparagine depletion by L-asparaginase in treating leukemia. Most leukemic cells lack the expression of ASNS, rendering themselves to rely on external asparagine (Asn) for growth and survival. L-asparaginase (ASNase) catalyzes asparagine conversion to aspartate, creating an Asn-depleted environment. Multiple signaling pathways are involved in the regulation of cellular response to asparagine depletion and therefore show synergy with ASNase when they are inhibited. Asn depletion directly activates the GCN2-ATF4 pathway to induce the expression of ASNS, driving de novo asparagine synthesis and cellular resistance to ASNase treatment. GCN2 inhibitors counteract GCN2-induced reduction in ASNS expression, restoring sensitivity to ASNase. In addition, GCN2 inhibitors alleviate GCN2-mediated repression of the stress MAPK pathway following ASNase treatment, promoting apoptosis. Downstream of the BTK pathway, MYC activates GCN2 through translation-dependent amino acid consumption and uncharged tRNA release. Thus, BTK inhibitors reduce MYC expression and suppress GCN2 activation following ASNase treatment. Asn depletion directly activates GSK3α, inducing GSK3-dependent protein degradation to release free amino acids. As a result, GSK3 inhibitors block this process and show synergy with ASNase treatment in leukemic cells.

**Figure 3 cancers-16-01171-f003:**
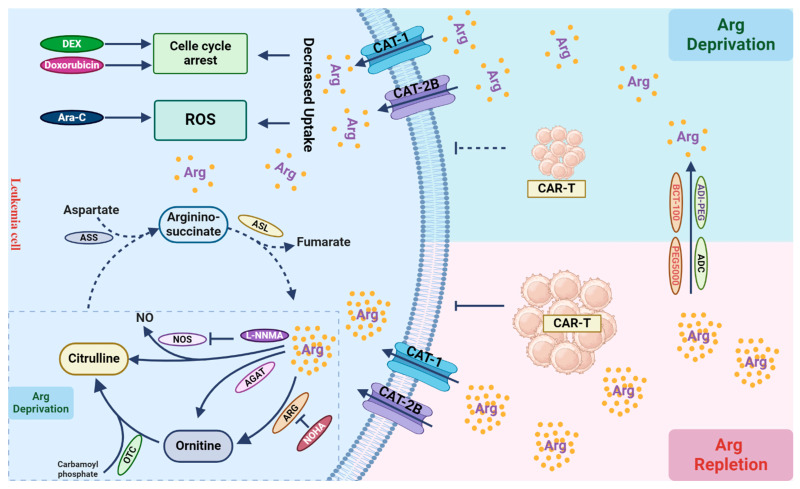
Arginine metabolism and its role in T cells. Leukemic cells primarily rely on arginine uptake for growth and proliferation. Arginine can be catabolized into ornithine within the urea cycle through the catalytic actions of arginase (ARG). Ornithine is then converted into citrulline by ornithine transcarbamylase (OTC). Due to the absence of argininosuccinate synthetase (ASS) and argininosuccinatelyase (ASL) in leukemic cells, they cannot autonomously synthesize arginine (dashed line). In tumor cells, arginine can also be enzymatically converted to nitric oxide (NO) and citrulline by nitric oxide synthase (NOS), or by arginine:glycine amidinotransferase (AGAT) into ornithine, leading to arginine depletion, although their activities in leukemic cells are not evident (dashed box). Arginine depletion by its catabolic enzymes leads to cell cycle arrest, increased cellular ROS production, and cell death, when combined with other chemotherapies, such as dexamethasone (DEX), doxorubicin, and cytarabine (Ara-C). Arginine depletion in the tumor microenvironment can suppress immune cell function, thereby compromising the effect of immunotherapy, such as CAR-T-based therapy. Inhibiting arginine catabolism in leukemic cells can restore arginine levels in the microenvironment, facilitating immune cell function to eradicate leukemic cells.

**Figure 4 cancers-16-01171-f004:**
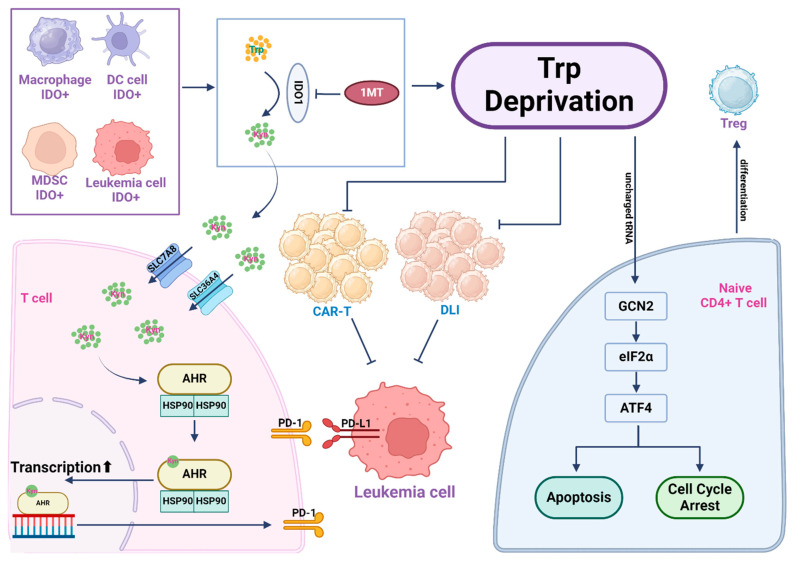
Tryptophan depletion and kynurenine pathway metabolites create an immunosuppressive microenvironment. Leukemic cells often exhibit elevated expression of indoleamine-2,3-dioxygenase 1 (IDO1), a key enzyme to drive tryptophan catabolism, resulting in extensive tryptophan consumption in the leukemia microenvironment. Within leukemic cells and several other IDO+ cells in the leukemia microenvironment, tryptophan is catabolized by IDO to yield kynurenine (Kyn), which can be released into the extracellular environment. T cells import Kyn via the transporter SLC7A8/SLC36A4. Kyn then binds to its nuclear receptor AhR to drive AhR-mediated gene transcription, which activates PD-1 expression on the surface of T cells and causes immunosuppression. T cells in a tryptophan-starved environment, influenced by uncharged Trp-tRNA, activate the GCN2-eIF2α-ATF4 pathway, leading to T cell apoptosis or cell cycle arrest. Activation of GCN2 in CD4+ T cells promotes Treg cell differentiation and activation. In cell therapy scenarios, due to tryptophan depletion, both CAR-T and Donor Leukocyte Infusion (DLI) functionalities are affected. In such instances, inhibiting IDO1 in the leukemic cells with 1-MT restores tryptophan levels in the microenvironment, enhancing the efficacy of cell-based immunotherapy.

**Table 1 cancers-16-01171-t001:** Ongoing and completed clinical trials involving perturbation of amino acid metabolism.

Registered Number	Disease	Phase	Group	Status	Reference
NCT02071927	R/R leukemia	I	Single-agent CB-839/CB-839 and AZA	Complete	N.A
NCT03047993	Advanced MDS	I/II	CB-839 and AZA	Complete	[29]
NCT04666649	R/R AML	I	Ven-PegC	Ongoing	N.A
NCT02899286	R/R AML	II	PEG-BCT-100	Complete	N.A
NCT01551628	R/R leukemia	I	Recombinant human arginase1 Peg-5000	Terminated(slow patient recruitment)	N.A
NCT02732184	R/R AML or MDS	II	Co-ArgI-PEG modified human arginase I	Complete	N.A
NCT01910012	R/RAML	II	ADI-PEG 20	Complete	[8]
NCT05001828	High risk AML	I	ADI-PEG 20, Venetoclax and Azacitidine	Ongoing	N.A
NCT02835729	ND-AML	I	Indoximodin, Idarubicin and Cytarabine	Complete	[30]
2006-005694-21 (EWALL-PH-01)	ND-Ph+ and/or CR-ABL1+ ALL	II	Dasatinib, cytarabine, asparaginase and methotrexate	Complete	[31]
NCT01085617(UKALL14)	newly diagnosed ALL	III	PEG-ASP and standard induction regimen (ph + disease received continuous oral imatinib)	Complete	[32]

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
