# Peer review of "Enhancing Leukemia Treatment: The Role of Combined Therapies Based on Amino Acid Starvation"

_cancers, 2024, doi:10.3390/cancers16061171_

Round 1

Reviewer 1 Report

Comments and Suggestions for Authors

In this review, the authors discuss the significance of amino acid deprivation in the treatment of various leukemias. Following a brief introduction, they present different strategies aimed at targeting Glutamine, Asparagine, Arginine, and Tryptophan. The authors effectively elucidate the rationale behind each strategy, the foundation of the studies, the diverse data available on the subject, and the results of each strategy in various pre-clinical and clinical studies. The text is very well written and documented. Additionally, they shed light on potential resistance mechanisms that cancer cells may develop and the impact on the immune system when pertinent. Overall, this review holds significant relevance in the field.

 I have only few remarks that need to be addressed before publication:

General remark 1: the abstract and the introduction are somehow redundant. The abstract could be more oriented by already citing the amino acids that will be discussed and to clearly state that some clinical trials are ongoing.

General remark 2: I believe that a table summarizing the ongoing clinical trials and the leukemia subsets will be helpful for the readers.

Line 124 : Can the authors explain the reason for FLT3-ITD targeting leads to defects in glutamine metabolism ?

Lines 149-154: Maybe it could be precised that the combination does not only affect the expression of MCL1. But more globally of cap-dependent translation

Line 175: first use of GSH please put full name

Figure1: The authors mention SLC7A1 whereas it is not cited in the text. Instead other transporters are mentioned (line 66). Either add a section in the text or adapt the figure that it fits with what is in the text.

Line 464-465: Please check this sentence. It is written here that tryptophan depletion induces AHR activation.

Author Response

In this review, the authors discuss the significance of amino acid deprivation in the treatment of various leukemias. Following a brief introduction, they present different strategies aimed at targeting Glutamine, Asparagine, Arginine, and Tryptophan. The authors effectively elucidate the rationale behind each strategy, the foundation of the studies, the diverse data available on the subject, and the results of each strategy in various pre-clinical and clinical studies. The text is very well written and documented. Additionally, they shed light on potential resistance mechanisms that cancer cells may develop and the impact on the immune system when pertinent. Overall, this review holds significant relevance in the field.

I have only few remarks that need to be addressed before publication:

General remark 1: the abstract and the introduction are somehow redundant. The abstract could be more oriented by already citing the amino acids that will be discussed and to clearly state that some clinical trials are ongoing.

Response: Thanks for your comment. We have modified the abstract to include the suggestion. Line 19~29.

General remark 2: I believe that a table summarizing the ongoing clinical trials and the leukemia subsets will be helpful for the readers.

Response: Aside from L-asparaginase, there are fewer ongoing clinical trials for combination therapies, as L-asparaginase trials are primarily focused on combination chemotherapy. Therefore, we have provided a table (Table 1, Line 144) to summarize the ongoing and complete clinical trials involving perturbing the metabolism of glutamine, asparagine, arginine, and tryptophan. We also clarified some of these trials in the text (highlighted in red).

Line 124 : Can the authors explain the reason for FLT3-ITD targeting leads to defects in glutamine metabolism?

Response: In the original text, it was observed that the uptake of glutamine decreases after the inhibition of FLT3-ITD, but the specific mechanism was not elaborated in detail. We have amended our text in Lines 125-128 to incorporate this opinion.

Lines 149-154: Maybe it could be precise that the combination does not only affect the expression of MCL1. But more globally of cap-dependent translation

Response: Thanks for your comment, we have amended the text in Lines 157-160.

Line 175: first use of GSH please put full name

Response: Thanks for your comment, we have defined the abbreviation in Line 60.

Figure1: The authors mention SLC7A1 whereas it is not cited in the text. Instead other transporters are mentioned (line 66). Either add a section in the text or adapt the figure that it fits with what is in the text.

Response: Thanks for your comment, we have revised Figure 1 and the legend (Line 194).

Line 464-465: Please check this sentence. It is written here that tryptophan depletion induces AHR activation.

Response: Thanks for your comment, we have revised it in Line 494-495.

Reviewer 2 Report

Comments and Suggestions for Authors

In this review authors provide a comprehensive description of amino acids particularly important in leukemia cells survival and targeting, providing also interesting information on the possible effect of amino acids deprivation during immunotherapy treatment.

Some minor comments and suggestions:

Line 38-41: add proper refs

Line 76: cycle twice

Refer to ABT-199 also as Venetoclax

Figure 2: specify that is a picture representing ALL

Provide a representation of Arginine metabolism and its role also in AML cells (to add at Fig3).

I suggest to create sub-chapters, for a better organization, where authors could list:  a) amino acid function b) amino acid targeting (or something similar).

Author Response

In this review authors provide a comprehensive description of amino acids particularly important in leukemia cells survival and targeting, providing also interesting information on the possible effect of amino acids deprivation during immunotherapy treatment.

Some minor comments and suggestions:

Line 38-41: add proper refs

Response: Thanks for your comment, we have incorporated additional references (Line 39~42).

Line 76: cycle twice

Response: Thanks for your comment, we have corrected it (Line 75).

Refer to ABT-199 also as Venetoclax

Response: Thanks for your comment, we have revised it (Line 108).

Figure 2: specify that is a picture representing ALL

Response: Thanks for your comment, we have revised it.

Provide a representation of Arginine metabolism and its role also in AML cells (to add at Fig3).

Response: Thanks for your comment, we have revised Figure 3 and the legend (Line 470-473)

I suggest to create sub-chapters, for a better organization, where authors could list:  a) amino acid function b) amino acid targeting (or something similar).

Response: Thanks for your comment, we have incorporated sub-chapters.